# Preparation and Performance of Mo/Cu/Fe Multi-Layer Composite Coating with Staggered Spatial Structure by Electro-Explosive Spraying Technology

**DOI:** 10.3390/ma15103552

**Published:** 2022-05-16

**Authors:** Kun Huang, Qiuzhi Song, Pengwan Chen, Ye Liu

**Affiliations:** State Key Laboratory of Explosion Science and Technology, Beijing Institute of Technology, Beijing 100081, China; hkhk1008@163.com (K.H.); pwchen@bit.edu.cn (P.C.); ly1989bit@163.com (Y.L.)

**Keywords:** electro-explosive spraying, multi-layer, Mo/Cu/Fe composite coating, staggered spatial structure

## Abstract

In the present study, electro-explosive spraying technology was used to prepare a multi-layer composite coating with a staggered spatial structure on a 45 steel substrate, and the mechanical properties and wear behavior of the coating were studied. The composite coating was prepared by spraying Mo as the bonding layer, then spraying high-carbon steel and aluminum bronze alternately as a functional coating. The cross-sectional morphology, surface morphology and the properties of the coating were analyzed with a scanning electron microscope (SEM), energy dispersive spectrometer (EDS), electron backscattered diffraction (EBSD) and a 3D profilometer. The bonding strength, friction and wear resistance of the coating were studied by the bonding strength experiment and by the friction and wear experiment. The results showed that it is feasible to prepare a composite coating with a sponge-like spatial structure with electro-explosive technology. There was metallurgical bonding as well as mechanical bonding between the adjacent coating layers. The composite coating had the advantages of uniform thickness, high compactness, high bonding strength and good wear resistance.

## 1. Introduction

With the rapid development of science and technology, more and more mechanical equipment needs to operate under harsh conditions such as high speeds and heavy loads, resulting in the increasingly prominent friction and wear problems of mechanical systems. In this type of equipment, the shaft–hole mechanical parts are widely used, such as various shafts and bearing bushes, pistons and cylinders in engines, etc. In order to improve the wear resistance of mechanical parts, the usual method is to process wear-resistant coatings on their working surfaces.

At present, the multi-component and multi-layer design methods are mainly adopted for the design methods of wear-resistant coatings. The multi-component design theories include: the multi-component alloying effect [1], the multi-phase composite synergistic effect [2], the nanocrystalline inlaid amorphous strengthening effect [3], the balance effect between the hard phase and the soft phase [4] and the friction reduction and wear resistance effect of tribochemical surfaces [5]. The multi-layer composite coating is constructed by multiple sub-layers designed using a multi-layer design method to obtain excellent wear resistance. The theoretical bases include the double affinity effect of the transition layer [6], the buffer effect of the gradient transition layer [7], the nano-multilayer grain refinement effect [8], the multi-layer interface dislocation motion pinning effect [9] and the inhibitory effect of alternating soft and hard cracks [10]. However, the multi-component design theory and the multi-layer design theory are mostly based on the assumption that the material structure is in an ideal state, and all kinds of effects depend on the actual structure of the coating [11,12,13,14].

The current thermal spraying technologies such as plasma spraying, high-speed flame spraying and laser cladding use external energy to heat the sprayed material to a molten state, and then it is sprayed onto the surface of the substrate at a higher speed. Firstly, the thermal radiation heats the substrate during the spraying process, which may change the performance of the substrate. Secondly, the sprayed droplets are prone to oxidation during the flight, resulting in oxide inclusions, which reduce the bonding strength of the coating. Finally, growth defects such as intergranular oxides and random voids are inevitably generated, which eventually leads to an unstable coating performance [15,16].

Electro-explosive spraying is a kind of technology which makes use of the explosion of a metal conductor when it is passed through a current with a high energy density in an instant. Partially molten metal droplets are sprayed to the surface of the substrate at a high speed by the shock wave generated during the explosion, and at the same time, they are rapidly condensed to form a coating [17,18]. The coating prepared by the technology is dense, the sprayed particles are fine and the particle size is uniform and adjustable. This technology can be used for spraying various surfaces, especially for the inner surface of the hole parts.

The advantages of electro-explosive spraying are that the spraying droplets have high temperatures and high speeds. When they impact the substrate, the metastable melting degree of the coating can be expanded, a metastable phase can be generated and segregation can be reduced, which is conducive to the formation of metallurgical bonding layers, and amorphous, microcrystalline and nanocrystalline structures can be formed [19,20,21,22]. The structures can significantly change the physical and mechanical properties of the coating material, such as changing the strength of the material, enhancing the wear resistance and improving the corrosion resistance. In addition, the performance of the substrate is not changed because the diameter of the metal droplets is micron, the interaction time with the substrate is extremely short and the heat-affected zone of the substrate is small.

Experiments show that the diameter of the metal droplets produced by electro-explosive spraying is related to the energy input to the metal wire [23]. At present, electro-explosive spraying technology is mainly used to prepare nano powders and coatings, such as titanium oxide, graphene nano sheets [23], TiC–TiB2–Ni composite coating [24], Al2O3-ZrO2 ceramic coating [25], Al-based amorphous composite coatings [26], submicron MoSi2-based coatings [27], molybdenum coatings [28], ceramic zirconium coatings [29], etc. At present, the coatings prepared by electro-explosive spraying are mainly a single layer stacked coating of a certain metal on the substrate, and research on composite coatings with a staggered spatial structure is very limited.

The aim of this study is to prepare a composite coating with a staggered spatial structure by means of the multi-component design method and the multi-layer design method using electro-explosive spraying technology, and to investigate the mechanical properties and wear resistance of the coating.

## 2. Coating System Design

In this paper, a Mo/Fe/Cu multi-layer composite coating was prepared on a 45 steel substrate by electro-explosive spraying technology, which combined the multi-component method and the multi-layer design method. The bases of the material selection were as follows: (1) Molybdenum is a high-melting-point metal, and its molten particle temperature is high, which could increase the interaction temperature between the particle and the substrate and could prolong the interaction time, and it is conducive to the formation of the metallurgical bonding layer and has a high bonding strength with most metals. Therefore, it is often used as a bonding layer in composite coatings [30]; (2) High-carbon steel is a commonly used steel tool with high-cost performance. When the high-carbon steel droplets are rapidly cooled from the molten state, a dense martensitic structure is formed with high hardness and high strength, and it is often used as a wear-resistant material [31,32,33,34]; and (3) aluminum bronze has good mechanical properties, a high thermal conductivity, good fatigue resistance and good wear reduction and wear resistance, so it is an ideal material for manufacturing high-load and high-speed bearings.

In this design, molybdenum was used as the bonding layer, and high-carbon steel and aluminum bronze were composited to form a 3D staggered spatial structure. The high-carbon steel with a high hardness mainly played the role of carrying the load in the coating, and the aluminum bronze with a soft hardness was used as a buffer and a lubricating material, which penetrated into the high-carbon steel coating with a spongy structure so that the two materials were fully interwoven and integrated into a whole. The composite coating of this structure had both a high load-carrying capacity similar to high-carbon steel and had self-lubricating properties similar to aluminum bronze, which was suitable for a variety of working occasions. This method can be used for the composite of other metallic materials, providing a reference for the design of new coating structures.

## 3. Experimental Equipment, Materials and Methods

### 3.1. Electro-Explosive Spraying Equipment

Self-developed electro-explosive spraying equipment was used in this experiment, which was mainly composed of a charging device, a capacitor energy storage device, a safety pulse discharge device and an explosion platform. The working principle is shown in Figure 1. The charging voltage range of this equipment was 5–50 kV, which was continuously adjustable, and the total capacitance of the charging capacitor bank was 112 μF. Before spraying, the capacitor energy storage device 3 was charged by the charging device composed of 1 and 2 so that it could store enough charges. During spraying, the air switch 4 was closed, and the capacitive energy storage device 3 was discharged to provide high-voltage electricity to the explosive chamber 6 to make the metal wire explode in an instant.

### 3.2. Spraying Materials and Spraying Parameters

The spraying materials were industrial metal wires with a diameter of 1.5 mm and a length of 140 mm. The wire materials were molybdenum, 80 high-carbon steel and aluminum bronze S216. The optimal explosive spraying voltage of each type of metal wire was determined by the comparison of many experiments under the existing equipment, which was 17.5 kV for the Mo wire, 14.5 kV for the steel wire and 15 kV for the aluminum bronze wire.

### 3.3. Preparation of the Coatings

Three types of samples were prepared for the experiments: (I) a 45 steel pipe with an inner diameter of 60 mm, an outer diameter of 70 mm and a length of 100 mm was used for the analysis of the thickness and cross-sectional morphology of the coating; (II) the samples with the size Ф25.4 mm × 7 mm were used for the bonding strength experiment; and (III) circular samples with inner diameters of 15.5 mm, outer diameters of 31.5 mm and a thickness of 10 mm were prepared for the wear-resistant experiments.

Before spraying, the samples were sandblasted then ultra-sonically cleaned with acetone for 5 minutes and then taken out to dry for later use. The installation methods of the samples in the electric explosive chamber during preparation are shown in Figure 2. The metal wire was parallel to the sample and was placed at the axis of the sample; then, the sample and the metal wire were fixed.

The electro-explosive spraying could be carried out when the sample and the wire were fixed and the spraying voltage corresponding to the metal wire was set. The spraying sequence of the composite coating was as follows: firstly, Mo was sprayed for 5 times as the bonding layer to improve the bonding strength of the coating and the substrate; then, two times, high-carbon steel, and two times, aluminum bronze were alternately sprayed as a group, and seven groups were continuously sprayed to form a functional coating. The total spraying times were 33. The three types of samples mentioned above were prepared one by one in this way. The preparation method of the high-carbon steel coating and aluminum bronze coating was the same, the substrate was 45 steel and the spraying times were 5 times for the molybdenum +28 times for the high-carbon steel and 5 times for the molybdenum +28 times for the aluminum bronze, respectively.

For the samples that had been sprayed, they were processed as follows before testing: (1) sample I was cut along the axial direction by wire cutting and divided into small sections with a length of 20 mm, cleaned with acetone, then inlaid and fixed, and then finally polished and etched with a 4% nitric acid alcohol solution for 10 s, then rinsed immediately with absolute ethanol, and dried for use; (2) sample II was sandblasted; and (3) the coated surface of sample III was ground and polished.

### 3.4. Testing Methods for Coating Properties

The coating thickness, cross-sectional morphology and elemental distributions of the coating were examined with scanning electron microscopy (SEM, Hitachi BCPCAS4800, Tokyo, Japan) coupled with energy dispersive spectrometer (EDS, Oxford X-Max 50, Oxford, UK).

A 3D profilometer (KEYENCE, VR-6000, Tokyo, Japan) was used to detect the surface of the composite coating, including the distribution of the different components and the porosity.

The electron backscattered diffraction (EBSD, JEOL, JSM-7900F, Tokyo, Japan) was used to analyze the coating grains. The sample preparation equipment was a vibration polishing machine (Buehler vibromet 2) and the sample was vibrated and polished using silica suspension with a particle size of 50 nm for 6 h.

A microcomputer-controlled electronic universal testing machine (Chenxin, WDW-E100D, Jinan, China) was used to test the bonding strength of the coatings. The tensile experiment was carried according to the requirements of the national standard GB/T8642-2002 “Determination of the Bonding Strength of Thermal Spray Coatings”. The experimental process was as follows: (1) the sandblasted sample II, FM-1000 adhesive and loading block were fixed on the clamping device; (2) the clamping device was put into the heating furnace, heated to 170–190 °C, kept for 3 h, removed when the power was cut off and the furnace temperature dropped to 50 °C and was cooled to room temperature; and (3) the clamping device was installed on the universal testing machine for a tensile test. The ambient temperature was room temperature, 25 °C, and the tensile loading speed was 0.001 mm/min. Four samples were tested for each experimental run and the bonding strength was given as an average of 4 measurements.

The friction coefficients were measured with universal friction and wear tester (Kesheng, MMW-1, Jinan, China) and the small face–face contact method was adopted. The surface of the sprayed sample III was ground and polished. The roughness (Ra) after polishing was 0.3–0.5 μm. The material of the paired sample was quenched 45 steel. The loads applied in the experiment were 100 N, 200 N, 300 N, 400 N, 500 N, 600 N, 700 N and 800 N, the relative sliding speed was 5.3 m/min, the time was 4000 s and lithium-based grease was used. Because the relative contact area between the sample and the paired sample was very small, the load applied to the sample was relatively large, and the friction performance under heavy load can better reflect the quality of the coating. The depth of the wear trace was measured using a 3D ultra-depth field microscope (KEYENCE, VHX-1000, Tokyo, Japan).

## 4. Results and Discussion

### 4.1. Thickness of the Coating

The SEM image of the cross-section of the coating is shown in Figure 3; the imaging mode was the secondary electron (SE). The thickness of the coating is the sum of the thickness of the bonding layer and the thickness of the functional coating, which is shown in Figure 3. The thickness of the coating of the whole sample was measured, and the minimum and maximum thickness at five different positions of the sample are listed in Table 1.

It can be seen from Table 1 that the average minimum thickness and maximum thickness of the coating was 194.6 μm and 236.4 μm, respectively, and that the standard deviation of the average values was 7.0 μm and 5.5 μm, respectively. The results show that there was a certain fluctuation in the coating thickness. This is because there was an error in the coating thickness formed after each solidification of the metal droplets, and the accumulated error after multiple spraying caused an uneven thickness. Because the size of the metal droplets was small and uniform, the coating was dense. By observing the thickness distribution of the entire sample along the axial direction, it was observed that: the middle part was thick, the two ends were thin and arc-shaped and the thickness difference was about 20 μm. The reason is that when the metal wire exploded, the metal droplets near the two ends were partially splashed out of the tubular sample and were not sprayed on the sample, resulting in less deposition at the two ends.

### 4.2. Bonding Strength of Coating

Bonding strength is one indicator for judging the coating quality. Figure 4 shows the tensile fractures of the composite coating after a tensile test. The cross-sectional metallographic image of the fracture section of the composite coating is shown in Figure 5.

Figure 4 and Figure 5 show that the fracture occurred at the bonding area between the bonding layer and the functional coating, indicating that the bonding strength between the functional layers (Cu/Fe) was greater than that between the functional coating and the bonding layer (Fe/Mo). The bonding strength data are shown in Table 2.

The experiment data in Table 2 show that the average bonding strength of the coating was 53.4 MPa and that the standard deviation was 1.8, indicating that the composite coating had a higher bonding strength.

### 4.3. Analysis of the Cross-Sectional Morphology of the Coating

EDS line scan of the cross-section of the composite coating is shown in Figure 6.

Figure 6 shows that there was obvious atomic diffusion between the adjacent coating layers in the composite coating, indicating that there was metallurgical bonding as well as mechanical bonding between the adjacent coating layers. Metallurgical bonding is beneficial to improve the bonding strength of composite coatings. By EDS line scanning of the bonding area shown in Figure 7, it can be seen that the iron in the substrate and the molybdenum in the bonding layer diffused each other with a depth of 1–1.5 μm, thus forming the metallurgical bonding.

It can be seen from Figure 6a that there was an obvious layered structure in the composite coating, that the bonding between the substrate and the coating and between the layers inside the coating was very tight and that there was no obvious void inside the coating. The reason is that in the process of electro-explosive spraying, the metal vapor vaporizing firstly removed the surrounding air under the impact of the explosion shock wave, forming a certain range of vacuum [19], and the subsequent molten metal droplets flew to the substrate at a high speed (1500–3000 m/s) [26] and rapidly spread and cooled, which formed a coating in an instant when they hit the substrate due to the low temperature of the substrate.

Some literature studies have shown that the maximum pressure of the explosion center can reach 100,000 atmospheres [20], the temperature can reach 10,000 °C and the droplet cooling rate can reach 10^7^ °C/s [20]. The sudden cooling of the metal can effectively prevent the growth of grains and can form amorphous, microcrystalline and nanocrystalline structures, which is beneficial to improve the hardness and other properties of the metal [25]. The composite mechanism of the electro-explosive coating is the repeated explosion and impact, which form a composite coating. Each new spray can strengthen the previously formed coating and can also fill the spreading gap left by the last spraying to form a mechanical riveting structure, which is also the main reason for the high bonding strength and compactness of the coating [19].

### 4.4. Analysis of Coating Surface Morphology

The morphology of the coating surface (polished and unetched) is shown in Figure 8. It can be seen from Figure 8a that on the same horizontal plane, various components are meshed and staggered, with aluminum bronze in the yellow area and the high-carbon steel in the gray area. The comparison diagram of the different components as shown in Figure 8b was obtained through the chromatic aberration calculation tool, in which the red area is the aluminum bronze and the black area is the high-carbon steel.

The area ratio of the aluminum bronze was calculated and is listed in Table 3. The calculation results show that the distribution of the aluminum bronze and high-carbon steel was uniform, and the area ratio of the aluminum bronze was about 53%, which is the same as the expected value of 50%, indicating that the electro-explosive spraying can well control the distribution of the different components in the composite coating.

Figure 9 is a comparison chart of the surface porosity of the coating. The porosity of the coating was calculated and is listed in Table 4.

The cross-sectional morphology of the coating detected by the metallographic microscope is shown in Figure 10. It can be seen from the figure that the structure of the composite coating was dense and different materials were distributed in wavy layers. Viewed from the substrate to the surface (from A to D), there was high-carbon steel and aluminum bronze on all heights (such as on the red line) and they were staggered and fused.

It can be seen from Figure 8 and Figure 10 that the high-carbon steel and aluminum bronze were interwoven in space in a mesh shape, with a similar volume ratio and uniform distribution, which conforms to the coating design scheme and verifies the feasibility of preparing the composite coating with a staggered spatial structure with electro-explosive spraying technology [27]. With the increase in spraying times, a sponge-like structure was gradually formed. The composite structure can give full play to the advantages of the two materials. The hard high-carbon steel can well support the external load [35], and the soft aluminum bronze has a self-lubricating ability, which can absorb a certain impact force and can also accommodate small hard particles from the outside to avoid scratches on the friction surface.

### 4.5. EBSD Analysis of the Composite Coating

The EBSD images of the composite coating are shown in Figure 11. As seen in the figure, the microstructures of the high-carbon steel in the composite coating are mainly austenite and ferrite, and the size of the austenite is refined to about 1 μm and the ferrite structure is refined to a nano level; after electro-explosive spraying, the grain of aluminum bronze was also refined to a micron level. The cooling rate of the metal droplets produced by the electric explosion was very large. The greater the undercooling, the higher the nucleation rate, and the lower the temperature of the new forming grains, which slows down the diffusion speed of atoms and makes the grains unable to grow sufficiently, so as to refine the grains. Grain refinement is beneficial to the improvement of the wear resistance of the coating [35,36]. Austenite and ferrite were formed at different stages of the droplet spreading. Austenite was the phase formed by the first spreading of the droplets. At this time, the droplet volume was large and the cooling speed was relatively slow, so the grain size was relatively large. Ferrite was formed by the secondary spreading of the splashed droplets after the droplets hit the substrate. Because the secondary splashed droplets were smaller and cooled faster, the formed grains were difficult to grow, resulting in the size of the ferrite grains to a nanoscale.

### 4.6. Wear Resistance Analysis

The samples of the high-carbon steel coating, composite coating and aluminum bronze coating after the tensile test are shown in Figure 12.

Five samples were tested for each type of coating under every load, and the coefficient of friction was given as an average of five values. The friction coefficients of each coating are shown in Figure 13. It can be seen that the friction coefficient of the composite coating was 6–18% lower than that of the high-carbon steel coating, and was significantly lower than that of the aluminum bronze coating, which was only about 1/4 of that of the aluminum bronze coating.

Under the loads of 700 N and 800 N, the aluminum bronze coating was worn and damaged, and the experiment could not be carried out. The reason is that the low hardness of the aluminum bronze led to the damage of the lubrication film on the coating under heavy load, resulting in adhesive wear, which quickly destroyed the coating structure, while the high-carbon steel coating and composite coating, due to the hard component support, could offset the pressure brought by heavy load.

The diagram of the friction coefficient with time under the load of 200 N is shown in Figure 14. It can be seen from Figure 14 that the whole friction and wear experiment had experienced two stages: the initial running-in stage and the stable-wear stage. During the initial cycles, the contact between the friction couples was actually the contact of the asperities; the grease did not fully play the role of lubrication at this time, so the friction coefficient was large. After many cycles, the asperities were gradually extruded, sheared and the contact surfaces were ground flat; the contact surfaces became smooth at this time, the grease was fully filled between the contact surfaces and fully played the role of lubrication, the friction coefficient decreased and the friction pair entered the stage of stable wear.

The worn surface morphologies of the composite coating are presented in Figure 15. Figure 15a shows that the maximum wear depth of the high-carbon steel coating was 6.496 μm, and there were many obvious line-like annular scratches on the surface of the coating, which were caused by the wear of the asperities between the paired sample and the coating. Figure 15b shows that the maximum wear depth of the composite coating was 15.69 μm, and there were relatively fine wear traces on the coating surface, indicating that the contact area between the paired sample and the coating surface was larger than that in the high-carbon steel coating experiment; the aluminum bronze in the composite coating had extruded traces, indicating that the aluminum bronze played the part of lubrication, so the friction coefficient of the composite coating was lower than that of the high-carbon steel coating. Figure 15c shows that the maximum wear depth of the aluminum bronze coating was 24.33 μm, which was the largest of the three coatings, and the wear was the most serious, indicating that the lubricating film formed by the grease was damaged in the experiment, resulting in direct contact between the paired sample and the coating. It can be seen from the wear morphology that the surface of the aluminum bronze had obvious adhesive wear, so the friction coefficient was the largest.

In addition, during the experiment, the sample with the high-carbon steel coating had slight vibrations and occasional noise, which may have been caused by the sudden increase in the friction coefficient due to the occasional oil film damage on the sample surface. In contrast, the sample with the composite coating was stable in the experiment without vibrations and noise.

Through the experimental analysis, it can be seen that the composite coating not only had the high bearing capacity of high-carbon steel, but also overcame the defect that the aluminum bronze coating could not work well under heavy load. The aluminum bronze in the composite coating could play its self-lubricating advantages, thereby reducing the friction coefficient of the composite coating and improving its lubricating performance. The experiments showed that the friction coefficient of the composite coating had little fluctuation, so the experimental process was stable and could meet the working requirements under various loads. The friction coefficient of the composite coating was the smallest, the friction resistance in operation was the smallest and the power loss was also the smallest.

## 5. Conclusions


It was feasible that the composite coating with a sponge-like spatial structure could be prepared by electro-explosive spraying technology. In the composite coating, high-carbon steel and aluminum bronze were closely combined and interwoven with each other in a staggered spatial structure, forming a sponge-like spatial structure.The EBSD analysis of the coating showed that the strengthening mechanism of the composite coating prepared by electro-explosive spraying was a fine-grain strengthening mechanism and that the minimum grain size could reach a nanometer level.The composite coating had high bonding strength. The tensile experiment showed that the minimum and maximum bonding strength of the composite coating were 51.1 MPa and 56.2 MPa, respectively, and the failure occurred in the bonding area between the bonding layer and the functional coating, indicating that the bonding strength between the functional layers was greater than that between the bonding layer and the functional coating. The coating had good compactness and the porosity of the coating was 0.42%.SEM and EDS analysis showed that there was obvious atomic diffusion between the adjacent coating layers in the composite coating, indicating that there was metallurgical bonding as well as mechanical bonding between the adjacent coating layers. Metallurgical bonding was beneficial to improve the bonding strength of the composite coating.The friction and wear experiments showed that the composite coating had a lower friction coefficient under a large load, which was 6–15% lower than that of the high-carbon steel coating, and only about 25% of that of the aluminum bronze coating, indicating that the composite coating had good wear resistance and that the aluminum bronze component in the composite coating played a role in lubricating and reducing wear.


## Figures and Tables

**Figure 1 materials-15-03552-f001:**
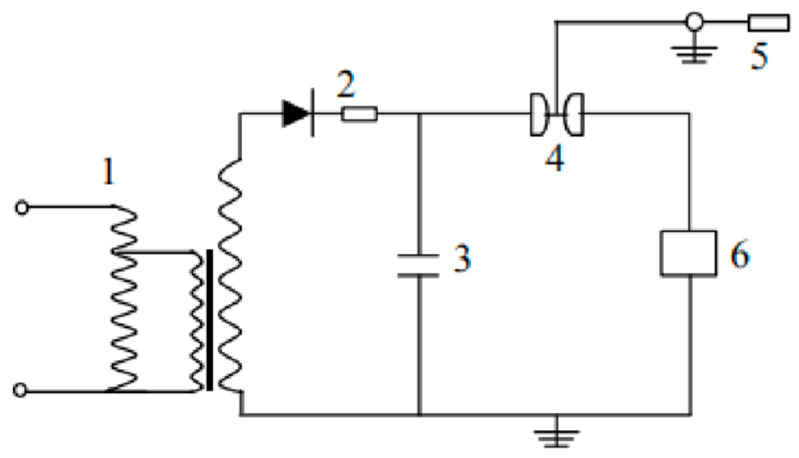
Schematic diagram of electro-explosive spraying experiment: 1—transformer; 2—ballast; 3—storage capacitor bank; 4—air switch; 5—grounding device; and 6—electric explosive chamber.

**Figure 2 materials-15-03552-f002:**
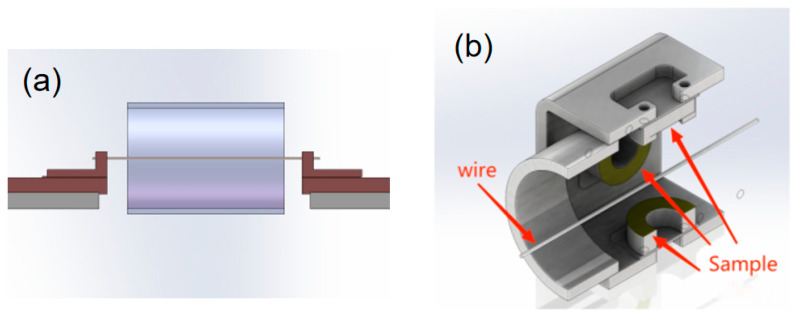
Schematic diagram of the installation methods of the samples: (**a**) sample I; (**b**) samples II and III.

**Figure 3 materials-15-03552-f003:**
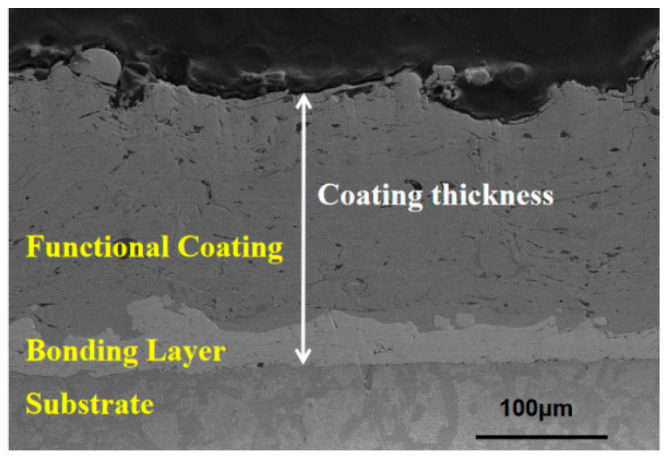
SEM image of coating cross-section (etched).

**Figure 4 materials-15-03552-f004:**
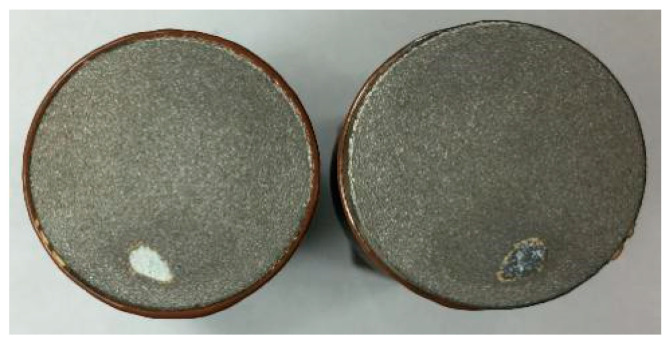
Photograph of the fracture section of the composite coating.

**Figure 5 materials-15-03552-f005:**
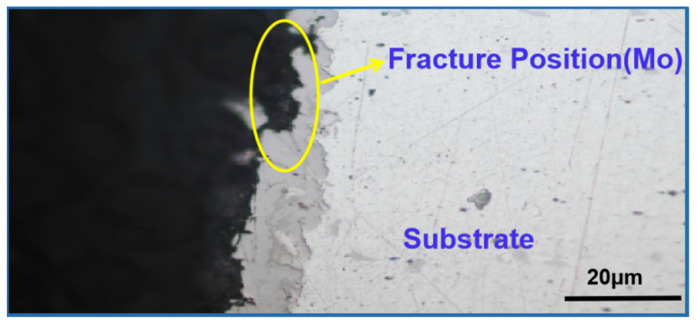
Cross-sectional metallographic image of the fracture section of the composite coating.

**Figure 6 materials-15-03552-f006:**
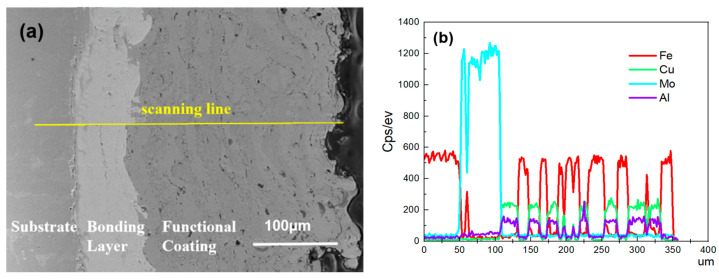
EDS line scan of the cross-section of the composite coating: (**a**) Cross-section; (**b**) Distribution chart of various elements.

**Figure 7 materials-15-03552-f007:**
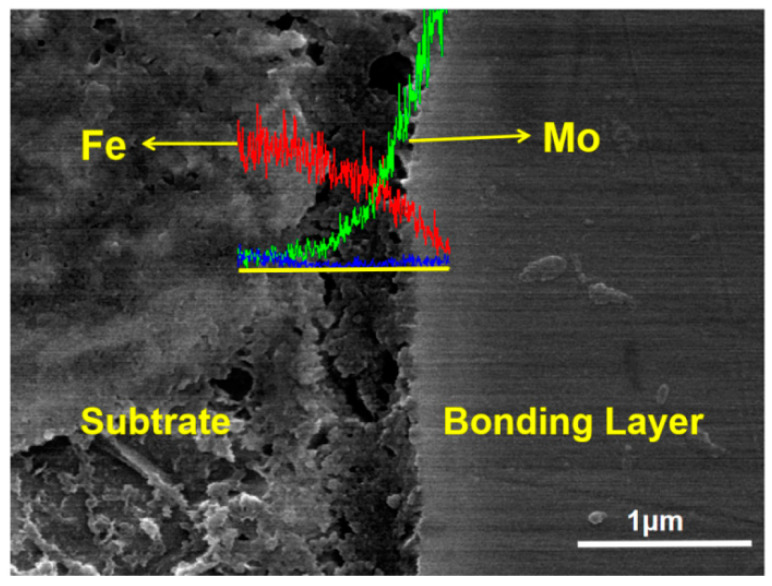
EDS line scan of bonding area between substrate and bonding layer.

**Figure 8 materials-15-03552-f008:**
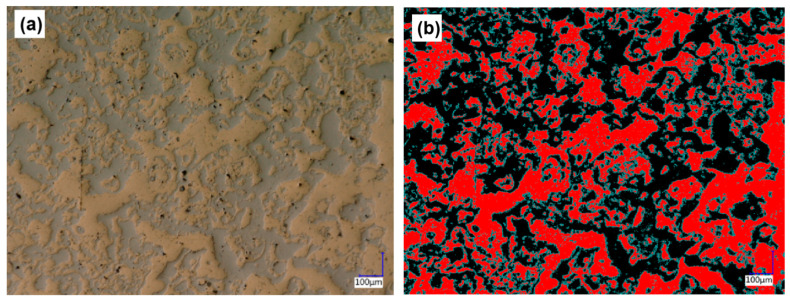
(**a**) 3D topography of the polished coating; (**b**) comparison chart of different components on the coating surface.

**Figure 9 materials-15-03552-f009:**
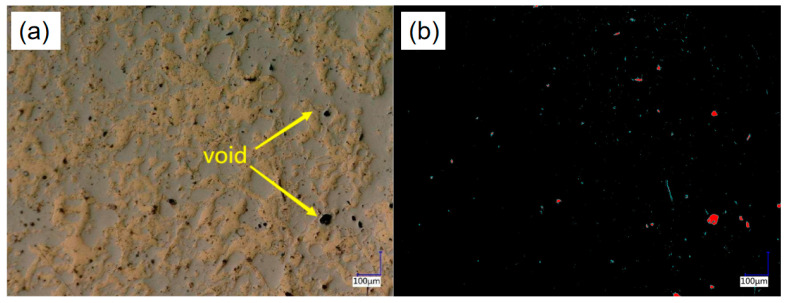
(**a**) 3D topography of the surface of the coating; (**b**) comparison chart of porosity.

**Figure 10 materials-15-03552-f010:**
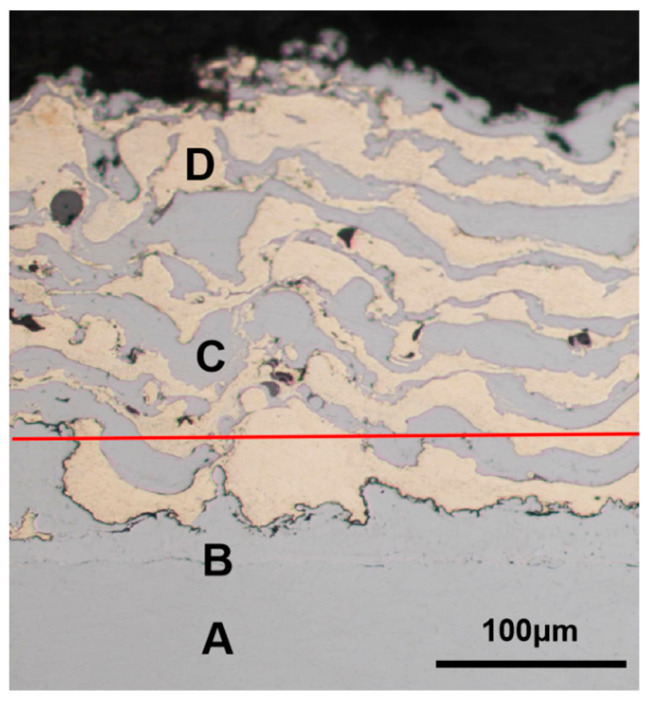
Cross-sectional view of composite coating: A—45 steel substrate; B—Mo layer; C—aluminum bronze layer; and D—high-carbon steel layer.

**Figure 11 materials-15-03552-f011:**
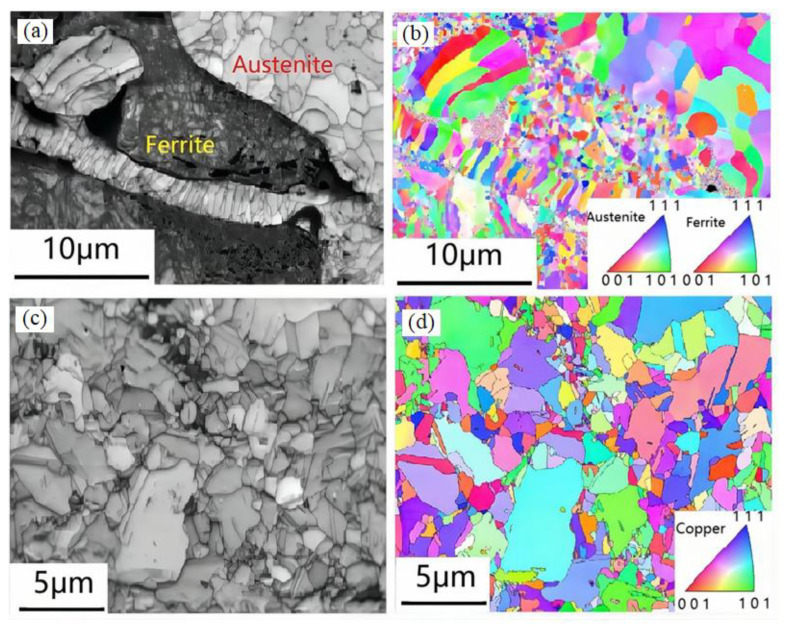
EBSD images of the composite coating: (**a**) IQ map of high-carbon steel area; (**b**) IPF map of high-carbon steel area; (**c**) IQ map of aluminum bronze area; and (**d**) IPF map of aluminum bronze area.

**Figure 12 materials-15-03552-f012:**
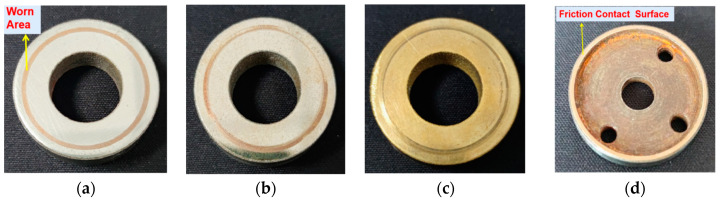
The samples after the friction and wear experiment: (**a**) sample with high-carbon steel coating; (**b**) sample with composite coating; (**c**) sample with aluminum bronze coating; (**d**) and paired sample in the experiment.

**Figure 13 materials-15-03552-f013:**
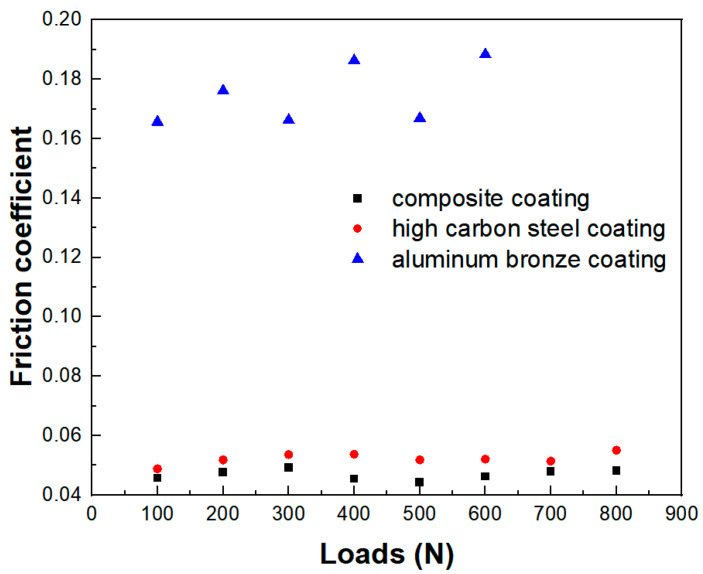
Friction coefficient diagram.

**Figure 14 materials-15-03552-f014:**
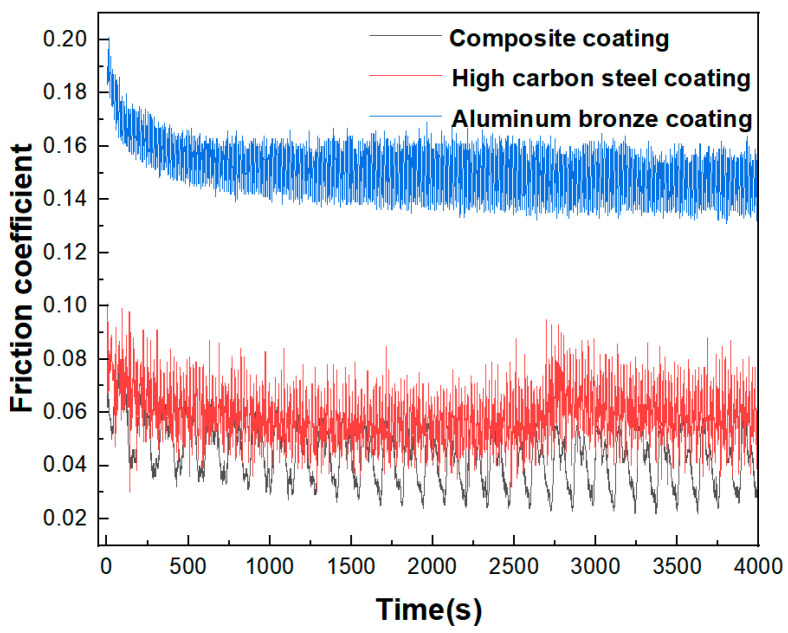
Diagram of friction coefficient with time under the load of 200 N.

**Figure 15 materials-15-03552-f015:**
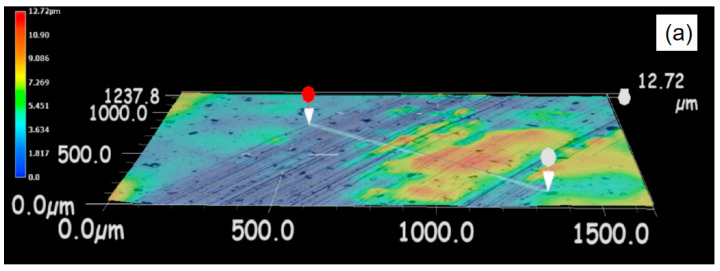
3D morphologies of the worn surface of the coatings: (**a**) high-carbon steel coating; (**b**) composite coating; and (**c**) aluminum bronze coating.

**Table 1 materials-15-03552-t001:** Coating thickness at different positions.

Position Number	1	2	3	4	5	Average Value/μm	Standard Deviation/μm
Thickness (min)	185	196	205	198	189	194.6	7.0
Thickness (max)	230	240	245	235	232	236.4	5.5

**Table 2 materials-15-03552-t002:** Bonding strength of the coating.

Sample Number	Bonding Strength/MPa	Remark
1	51.1	Sample II
2	53.8	Sample II
3	56.2	Sample II
4	53.1	Sample II

**Table 3 materials-15-03552-t003:** Calculation of area ratio of aluminum bronze.

Item	Value
Aluminum bronze area/μm^2^	1,129,502
Counting point	17,603
Whole area of the region/μm^2^	2,146,703
Area ratio of aluminum bronze/%	53

**Table 4 materials-15-03552-t004:** Calculation of porosity of the coating.

Item	Value
Pore area/μm^2^	9173
Counting point	1275
Whole area of the region/μm^2^	2,146,703
Porosity/%	0.42

## Data Availability

Not applicable.

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
