# Peer review of "Preparation and Performance of Mo/Cu/Fe Multi-Layer Composite Coating with Staggered Spatial Structure by Electro-Explosive Spraying Technology"

_materials, 2022, doi:10.3390/ma15103552_

Round 1

Reviewer 1 Report

The comments are mentioned in the attached file.

Author Response

Dear Editors and Reviewers:

We appreciate you and the reviewers for your precious time in reviewing our paper and providing valuable comments. Those comments are all valuable and very helpful for revising and improving our paper, as well as the important guiding significance to our researches. The authors have studied the comments carefully. We agree with all reviewers and have tried our best to make correction. We hope the manuscript after careful revisions meet your hign standards. The authors welcome  further constructive comments if any. Revised portion are marked in the revised manuscript (track changes marked).

Below we provide the point by point responses. The editor/reviewers’ comments are shown in Italic, our response is in blue.

Reviewer 2 Report

Please rewrite the artcle,  currently methodology is mixed with results and discussion. No wear resistance results other than coefficient of friction, no analysis of samples after wear test. Please add. Minor comments in attached file 

Author Response

(The authors gave the same response as above.)

Reviewer 3 Report

The present work deals with the Mo/Cu/Fe multi-layer composite coating prepared by electro-explosive spraying technology. The manuscript is well written, however, some major revisions are mandatory before it can be recommended for the publication. The main comments are given below.

  1. In Introduction and Abstract, the motivation / idea / aim of the work is missing and should be added.
  2. In Introduction, the overview related to electro-explosive spraying technology should include references with various materials used for coatings, materials of substrates, etc.
  3. Introduction is written in a book writing style. Moreover, the use of future tense is not appropriate.
  4. The “Coating system design” chapter contains the information that should be referenced. The same is valid for lines 235-237 and 277-280.
  5. Lines 57-58: The sentence is not clear. What did authors mean by “small change in technological parameters and operations”?
  6. Figs. 2, 3b and 5a are redundant, since they do not bring any relevant information. The schemes are sufficient.
  7. Authors mention three different samples – I, II and III. What was the preparation procedure of the samples II and III?
  8. Please, specify the imaging mode of SEM used in Fig. 4.
  9. Table 1 should be interconnected with Fig. 4 (or relevant SEM image). At least some of the positions, where the thickness of coating was measured, could be highlighted in the SEM image. Moreover, the standard deviations for average values of thickness are missing in Table 1.
  10. Authors mention minimum and maximum values of thickness. Is the minimum / maximum thickness of coating related to the thickness of coating without / with the bonding layer, respectively? Please, describe in more details.
  11. The authors claim that the surface of the coating is relatively smooth, but compared to the surface of the substrate, the exact opposite appears, also with respect to the scale given in the SEM image in Fig. 4. Please, give a statement.
  12. If the sentence “By observing the thickness distribution of the entire sample along the axial direction, it can be seen that: …” is not referenced to Fig. 4, the term “it can be seen” should be rewritten, to the other form, for example: “it has been observed that”.
  13. Text between lines 176 and 192 should be moved to Experimental equipment, materials and methods. The same is valid for lines 211-213, 285-287 and 292-298.
  14. In Fig. 9, the scanning line should be highlighted.
  15. Authors claim that the fracture occurs at the bonding area between the bonding layer and the substrate. However, Fig. 7 shows that the interface between the substrate and the bonding layer is intact, while the interface between the bonding layer and the functional layer is broken. Please, give a statement.
  16. The term “corroded” is confusing. Instead, authors could use the term “etched” in the whole manuscript.
  17. The measured values in Fig. 13 should not be interconnected with solid lines. If authors want to show a trend, they should use an approximation function.
  18. What is the authors’ opinion on relatively wide variance of friction coefficient values of aluminum bronze coating at different loads?
  19. Microstructural and phase analyses are missing in the manuscript. These results could significantly contribute to better understanding of properties. Please, give a statement.

Besides the above mentioned comments, the manuscript needs some other minor editing:

- the document formatting should be improved (for examples spacing, consistency of references, etc.)

- line 31: need ---> needs

- some sentences should be rewritten (e.g. lines 37-38, 42-43, 47-48, 193-194), the same is valid for figure captions (Figs. 6 and 7) 

- line 100: metal materials ---> metallic materials

- line 107: was shown ---> is shown

- line 153: (3) ---> (2)

- line 166: shows ---> show

- line 202: New paragraph is not necessary.

- Table 2: designation of the sample (sample II) could be added into the table as a part of “Remark” column.

- Fig. 8a: “Functional” should be written in a single line.

- line 205: MP ---> MPa; standard deviation should also be added.

- line 213: “:” ---> “.”

- line 236: To increase the scientific level, SI units should be used.

Author Response

(The authors gave the same response as above.)

Round 2

Reviewer 1 Report

The authors answered to my suggestions.

Reviewer 2 Report

Thank you for the answers

Reviewer 3 Report

Authors improved the quality of the manuscript, however, I have few more comments which could be reflected in the manuscript:

- the EBSD experiment seems to be incomplete - I would expect an information related to the orientation of grains (e.g. 2D EBSD color maps with orientational scale). Authors showed the microstructural images only.

- line 420: the standard deviation value should have the same number of decimal places as the average value

- I would recommend using "μm" rather than "microns".

After the minor revision, I can recommend the manuscript to be published.
